# DSP: Dynamic Sequence Parallelism for Multi-Dimensional Transformers

Xuanlei Zhao [1]   Shenggan Cheng [1]   Chang Chen [1]   Zangwei Zheng [1]   Ziming Liu [1]   Zheming Yang [1]   Yang You [1]

## Abstract

Scaling multi-dimensional transformers to long sequences is important across various domains. The challenges of large memory requirements and slow speed of such sequences require sequence parallelism. All existing approaches fall under the category of embedded sequence parallelism, which are limited to shard along a single sequence dimension, thereby introducing significant communication overhead. However, multi-dimensional transformers involve independent calculation across multiple sequence dimensions. To this end, we propose Dynamic Sequence Parallelism (DSP) as a novel abstraction of sequence parallelism. DSP dynamically switches the parallel dimension according to the computation stage with efficient resharding strategy. DSP offers *significant reductions in communication costs, adaptability across modules, and ease of use with minimal constraints*. Experiments demonstrate DSP's superiority over state-of-the-art sequence parallelism methods by remarkable throughput improvements ranging from 32.2% to 10×, with at least 50% communication volume reduction.

## 1. Introduction

Efficiently scaling multi-dimensional transformers to accommodate long sequences is necessary across diverse domains, including video generation (Singer et al., 2022; Blattmann et al., 2023; Ma et al., 2024), image generation (Ramesh et al., 2021; Rombach et al., 2022; Liu et al., 2024), protein structure prediction (Jumper et al., 2021), spatial-temporal information processing (Cong et al., 2021), and beyond. The long length of sequences introduces substantial activation memory costs and notable slowdown for speed, underscoring the need for employing parallelism.

Apart from data parallel and pipeline parallel (Huang et al., 2019) which cannot reduce memory cost and inference time,

sequence parallel is the only option. Current sequence parallelism, such as Megatron-LM (Shoeybi et al., 2019), Ring-Attention (Li et al., 2021; Liu et al., 2023a), Megatron-SP (Korthikanti et al., 2022), and DeepSpeed-Ulysses (Jacobs et al., 2023) are all embedded sequence parallelism methods. As shown in Figure 1, these embedded methods shard along a single sequence dimension, which are tailored to the specific pattern and introduce extra communication and complex code modification.

However, multi-dimensional transformers calculate independently across multiple sequence dimensions. For instance, for video generation models like OpenSora (Zangwei Zheng, 2024) and Latte (Ma et al., 2024), Spatial-Temporal Attention (Yan et al., 2021) is adopted which separates attention computations to independent temporal and spatial computation. Therefore, there exists a potential space for a new sequence parallelism paradigm.

To adapt to the flexible patterns of multi-dimensional transformers, we introduce Dynamic Sequence Parallelism (DSP) as a novel abstraction of sequence parallelism, featured by its *elegant design, high effectiveness, and excellent compatibility*. Unlike embedded sequence parallelism, DSP dynamically switches the parallel dimension of sequences during the computation stage with an efficient resharding strategy, completely decoupled from the modules' logic.

DSP offers several advantages over embedded sequence parallelism: 1) *Efficient communication*: DSP incurs significantly lower communication costs due to its simplified communication patterns and reduced frequency of exchanges. 2) *Adaptability*: DSP seamlessly adapts to most modules without necessitating specific modifications and imposes few limitations on its usage. 3) *Ease of use*: DSP is remarkably easy to implement, and also provides a simple API for users to enable it effortlessly.

Our experiments yield promising results, showcasing DSP's superiority over state-of-the-art embedded sequence parallelism methods. It achieves an end-to-end throughput improvement ranging from 32.2% to 10× and reduces communication volume by at least 75%.

We summarize our contributions as follows:

- We introduce DSP as a novel abstraction of sequence parallelism aimed at effectively scaling multi-

---

[1]National University of Singapore. Correspondence to: Yang You <youy@comp.nus.edu.sg>.

*Proceedings of the 42nd International Conference on Machine Learning*, Vancouver, Canada. PMLR 267, 2025. Copyright 2025 by the author(s).

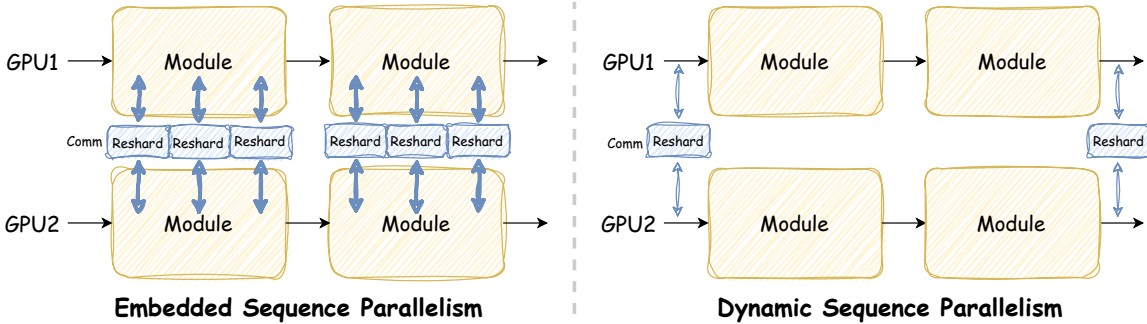

*Figure 1.* Comparison of Embedded and Dynamic Sequence Parallelism. Reshard means the communication to change sequence parallel layout. The blue arrow represents communication. The number and width of the arrows indicate the volume and frequency of communication, respectively.

dimensional transformers. DSP dynamically switches the parallel dimension of sequences during the computation stage, offering high effectiveness, elegant formalism, and excellent compatibility.

- By significantly reducing communication volume and frequency, DSP improves end-to-end throughput by 32.2% to 10× and reduces communication volume by at least 50% compared to state-of-the-art methods.

- DSP seamlessly integrates with various modifications without requiring specific modifications and imposes few limitations. Its ease of use is highlighted by the minimal code changes needed to incorporate it into existing frameworks with our high-level API.

## 2. Related Work

*Table 1.* Meanings of the symbols that are used in this paper.

| | |
|---|---|
| $B$ | The number of batch sizes. |
| $C$ | The size of hidden states. |
| $M$ | The volume of a sequence tensor. |
| $N$ | The number of GPUs. |
| $n$ | The $n$-th GPU. |
| $\mathbf{X}$ | The a multi-dimensional sequence. |
| $\mathbf{X}_{p,n}$ | The partition of X assigned to GPU $n$. |
| $S_i$ | The $i$-th sequence dimension. |
| $s_i$ | The status that sequence is sharded from dimension $S_i$. |
| $\hat{s}$ | The status that sequence is not sharded. |

### 2.1. Background

**Transformer Architecture.** Transformer (Vaswani et al., 2017) is a type of neural network architecture that has become highly influential in natural language processing (Devlin et al., 2018; Brown et al., 2020; Reid et al., 2024) and other domains (Dosovitskiy et al., 2020; Jumper et al., 2021;

Peebles & Xie, 2022). The Transformer is composed of a stack of layers, each consisting of a multi-head attention (MHA) and a position-wise feed-forward network (FFN). Specifically, the MHA comprises H independently parameterized attention heads, formulated as:

$$\text{MHA}(x) = \text{Concat}(\text{head}_1, \ldots, \text{head}_H)\mathbf{W}^O, \quad (1)$$

$$\text{head}_i = \text{Att}(\mathbf{Q}_i, \mathbf{K}_i, \mathbf{V}_i). \quad (2)$$

$$\text{Att}(\mathbf{Q}, \mathbf{K}, \mathbf{V}) = \text{softmax}\left(\frac{\mathbf{Q}\mathbf{K}^\top}{\sqrt{d_k}}\right)\mathbf{V}, \quad (3)$$

$$x_{\text{MHA}} = \text{LayerNorm}(x + \text{MHA}(x)). \quad (4)$$

where $\text{Att}(\cdot)$ denotes the scaled dot-product attention, $\mathbf{Q}, \mathbf{K}, \mathbf{V}$ are query, key, value projections, and LayerNorm is the layer normalization. The output $x_{\text{MHA}}$ is fed into the FFN, which consists of two linear transformations with a ReLU activation in between, computed as:

$$\text{FFN}(x) = \max(0, x\mathbf{W}_1 + \mathbf{b}_1)\mathbf{W}_2 + \mathbf{b}_2, \quad (5)$$

$$x_{\text{out}} = \text{LayerNorm}(x_{\text{MHA}} + \text{FFN}(x_{\text{MHA}})), \quad (6)$$

where $\mathbf{W}_1, \mathbf{W}_2, \mathbf{b}_1, \mathbf{b}_2$ are the parameters of the FFN.

**Multi-Dimensional Transformer.** Multi-dimensional transformers (Ho et al., 2019; Yang et al., 2022) extend the self-attention mechanism of standard transformers to operate over multiple dimensions beyond just one sequence. An example of 2D-Transformer is shown in Figure 2. Let the input multi-dimensional sequence be denoted as $\mathbf{X} \in \mathbb{R}^{[B, S_1, S_2, \ldots, S_K, C]}$, where $B$ is the batch size, $S_1, S_2, \ldots, S_K$ are the sequence lengths along $K$ different sequence dimensions, and $C$ is the hidden size. Multi-dimensional transformer can be formatted as:

$$\mathbf{X}_{\text{reshape}} = \text{Reshape}(\mathbf{X}, [B \times \prod_{j \neq i} S_j, S_i, C]). \quad (7)$$

The transformer block operation is then applied along the $i$-th sequence dimension of $\mathbf{X}_{\text{reshape}}$.

$$\mathbf{X}_{\text{out}} = \text{transformer\_block}(\mathbf{X}_{\text{reshape}}). \quad (8)$$

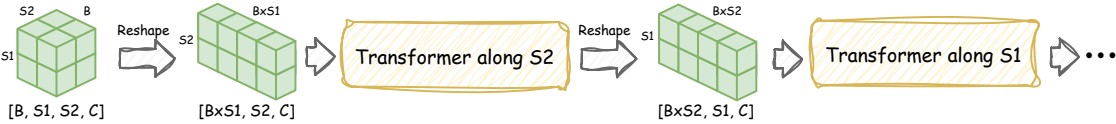

*Figure 2.* Illustration of multi-dimensional (2D for this example) transformer. It calculates each dimension of sequence independently at the corresponding stage. After the calculation is done in one dimension, it will switch to another dimension in next stage.

After applying the transformer block operation along all $N$ dimensions, the final output tensor $\mathbf{X}_{\text{out}}$ has the same shape as the input tensor $\mathbf{X}$. Multi-dimensional Transformer is widely used for applications with multi-dimensional inputs including video data (Xu et al., 2020; He et al., 2021; Geng et al., 2022; Ma et al., 2024), 3D data (Zheng et al., 2021; Chen et al., 2023), protein structure prediction (Jumper et al., 2021; Mirdita et al., 2022), time-series data (Pan et al., 2022; Huang et al., 2022; Deihim et al., 2023) and beyond.

## 2.2. Related Work

In this section, we discuss the four main parallelism techniques employed in deep learning: data parallelism, tensor parallelism, pipeline parallelism, and sequence parallelism.

Data parallelism (Hillis & Steele Jr, 1986; Li et al., 2020) is one of the most widely adopted parallelism techniques. The input data is partitioned across devices, each processing a subset. Model parameters are replicated, and gradients are summed. ZeRO (Rajbhandari et al., 2019; 2021) optimizes memory by partitioning parameters, states, and gradients across devices, enabling the training of larger models. Tensor parallelism (Shazeer et al., 2018; Shoeybi et al., 2019), or model parallelism, partitions model parameters across devices. Different model parts are assigned to different devices. Pipeline parallelism (Huang et al., 2019; Narayanan et al., 2019; Li & Hoefler, 2021; Liu et al., 2023b) partitions the model into stages executed in parallel across devices. Activations are passed between devices in a pipeline style.

Unlike parameter parallelism discussed earlier, sequence parallelism is a technique specifically designed for distributing long sequences and activation across multiple devices. Here are three main methods of sequence parallelism:

**Ring Attention.** Li et al. (2021) employs an innovative approach to partitioning the sequence dimension using a ring-style peer-to-peer (P2P) communication pattern to transfer keys and values across GPUs. (Liu et al., 2023a) enhance it with an online softmax mechanism, allowing for the computation of attention scores without retaining the full sequence length. However, Ring Attention's reliance on P2P communication can be less efficient in high-latency environments.

**Megatron-LM.** Shoeybi et al. (2019) introduces tensor tensor parallelism by partitioning model across devices. The method splits both feed-forward networks and self-attention layers along their hidden dimension, with necessary all-reduce operations. This approach enables efficient distributed training while reducing memory and communication overhead compared to data parallelism. It also reduces the tensor size on each devices.

**Megatron-SP.** Korthikanti et al. (2022) further optimizes activation usage in the attention based on tensor parallelism. To transit between tensor parallelism and sequence parallelism in the transformer block, additional all-gather and reduce-scatter operations are introduced. But it's limited by the number of attention heads, as self-attention relies on the parallelism of the head dimension of the sequence.

**DeepSpeed-Ulysses.** Jacobs et al. (2023) introduces an innovative approach for training long sequences by utilizing all-to-all collective communication. This method partitions the query, key, and value matrices across attention heads while preserving the original attention computation structure. The process is facilitated by two sets of all-to-all communications that alternate between sequence splitting and attention head splitting. Nevertheless, it is constrained by the number of attention heads as well.

Moreover, these sequence parallelism methods are designed for parallelism within a single sequence dimension. For multi-dimensional transformers, this strategy becomes inefficient due to unnecessary communication. While specialized parallelism for multi-dimensional sequences has been explored in specific domains (Cheng et al., 2024), their applicability remains limited.

## 3. Dynamic Sequence Parallel

### 3.1. Problem Definition

In sequence parallelism, the objective is to distribute activation computations across multiple GPUs to reduce the memory overhead caused by long sequences. This approach, however, incurs additional communication costs between GPUs. Our goal is to optimize this trade-off in the context of multi-dimensional transformers.

Given a sequence $\mathbf{X} \in \mathbb{R}^{[B,S_1,S_2,...,S_K,C]}$ and a set of $N$ GPUs, where $S_1, \ldots, S_K$ are the sequence along $K$ different sequence dimensions, we aim to partition the computation such that the memory usage per GPU is under capacity while maintaining acceptable communication costs. Let

*Table 2.* Definition of Dynamic Primitives for DSP. $s_i$ denotes the sequence sharded from dimension $i$, while $\hat{s}$ indicates the sequence is not sharded. $M$ represents the sequence size, and $N$ signifies the sequence parallel size.

| Source Shard | Target Shard | Primitives | Comm Operation | Comm Volume | Freq |
|:---:|:---:|:---:|:---:|:---:|:---:|
| $s_i$ | $s_i$ | / | / | / | / |
| $s_i$ | $s_j$ | Switch | all-to-all | $M/N$ | High |
| $\hat{s}$ | $s_i$ | Split | / | 0 | Low |
| $s_i$ | $\hat{s}$ | Gather | all-gather | $M$ | Low |

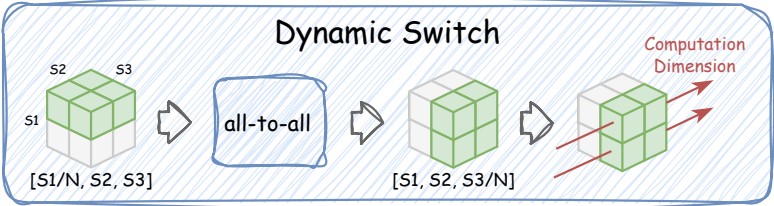

*Figure 3.* Illustration of dynamic switch. Through all-to-all communication, the computation sequence is

$\mathbf{X}_{p,n}$ denote the partition of $\mathbf{X}$ assigned to GPU $n$, where $p$ represents the partition strategy. It can be formulated as:

$$\min_{p} \sum_{n=1}^{N} \mathrm{CommCost}(\mathbf{X}_{p,n}),$$

$$s.t. \ \mathrm{Memory}(\mathbf{X}_{p,n}) < Capacity. \qquad (9)$$

where $\mathrm{Memory}(\mathbf{X}p, n)$ denotes the memory usage of partition $\mathbf{X}_{p,n}$ on GPU $n$, $\mathrm{CommCost}(\mathbf{X}_{p,n})$ represents the communication cost. We aim to achieve a balance that minimizes the overall computational overhead while optimizing GPU resource utilization.

### 3.2. Dynamic Primitives

The key dynamic primitives of DSP are outlined in Table 2. These three primitives form the foundation for adapting DSP to various multi-dimensional transformers.

The first condition is that when there is no need to alter sequence parallelism between computation stages, we maintain the shard status of the sequence. This approach significantly reduces unnecessary communication overhead. However, when it becomes necessary to transit parallelism between dimensions, we employ dynamic switch to efficiently transform parallelism. Specifically, as depicted in Figure 3, dynamic switching adjusts the parallelism to a dimension unrelated to the ongoing computation, utilizing highly efficient all-to-all operations.

Assume $\mathbf{X} \in \mathbb{R}^{[B,S_1,...,S_i/N,...,S_j,...,S_K,C]}$ represents the input. The current parallel dimension is $i$ so its sequence length is $S_i/N$ on each device, where $N$ is sequence parallel size. If we want to switch the shard dimension from $i$ to $j$,

the operation can be formulated as follows:

$$\mathbf{Y} = \mathrm{DynamicSwitch}(\mathbf{X}, i, j), \qquad (10)$$

where $\mathbf{Y}$ has the shape $\mathbb{R}^{[B,S_1,...,S_i,...,S_j/N,...,S_K,C]}$. Furthermore, Split and Gather operations facilitate smooth transitions between sharded and non-sharded states. Although these operations may involve increased communication compared to Switch operations, they are primarily utilized at the onset and conclusion of most networks, and also for some global operations in very rare conditions, rendering their costs negligible.

### 3.3. Overview

In the realm of multi-dimensional transformers, computation occurs independently for each sequence dimension. To harness this inherent feature, we introduce Dynamic Sequence Parallelism (DSP), an efficient, adaptive and ease-of-use method for multi-dimensional transformers.

To ensure correct computation logic with sequence parallelism, embedded methods typically require complex and time-consuming communications within computation modules to change the parallel dimension. As illustrated in Figure 4, the key feature of DSP is its dynamic switch of parallel dimension between computation stages.

By resharding only between computation stages dynamically, rather than within them, this approach allows DSP to remain independent of the computation logic within the module. Therefore, DSP eliminates numerous unnecessary communications within modules, and is able to utilize efficient all-to-all operations to switch parallelism dimensions for the intermediate sequence. For operations involving all

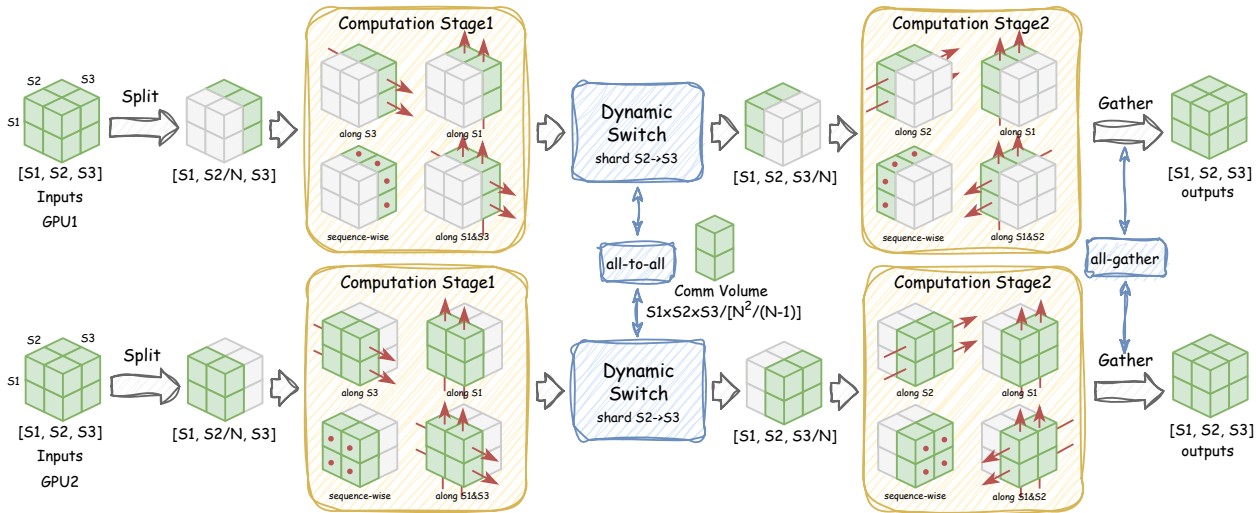

*Figure 4.* System overview of Dynamic Sequence Parallelism. It utilizes *Split* and *Gather* at the beginning and the end of model to separate and collect the complete sequences. In the middle computation, it utilizes *Dynamic Switch* to change the sharding dimension with efficient all-to-all communication. So that the following computation will not be affected by sharding.

sequence dimensions, including the beginning and end of the model, DSP handles them by Split and Gather operation.

Furthermore, we also propose a high-level, user-friendly implementation of DSP compatible with all distributed frameworks based on PyTorch.

### 3.4. Adaptability and Flexibility

Given its decoupling from the computation of modules, DSP exhibits remarkable adaptability, making it compatible with a wide array of transformer variants such as Cross Attention (Hertz et al., 2022; Ma et al., 2024); specialized kernels like FlashAttention (Dao et al., 2022); special attention mechanisms including multi-query attention (Shazeer, 2019) and grouped-query attention (Ainslie et al., 2023); and even beyond like Mamba (Gu & Dao, 2023) and RWKV (Peng et al., 2023). This inherent flexibility enables DSP to seamlessly integrate into diverse transformers without specific modification. Furthermore, while DeepSpeed-Ulysses and Megatron-SP necessitate attention head splitting, DSP's scalability is significantly better because it shards on sequence length, which is more redundant.

Moreover, DSP's adaptability extends beyond module compatibility to encompass various parallelism methodologies. From conventional data parallelism to more sophisticated approaches like ZeRO and pipeline parallelism, DSP effortlessly integrates with diverse parallel computing paradigms, thereby enhancing scalability and performance across distributed computing environments.

By calling just four functions without knowing the detailed implementation, DSP can be enabled on PyTorch and is compatible with various distributed frameworks, including FSDP (Zhao et al., 2023), Accelerate (Gugger et al., 2022), DeepSpeed (Rasley et al., 2020), and Megatron-LM (Shoeybi et al., 2019).

## 4. Theoretical Analysis

We choose 2D-Transformer as described in Equation 8 as our base model, which is widely employed in real-world applications. To be specific, we use the Open-Sora (Zangwei Zheng, 2024) variant of 2D-Transformer, an open-source video generation model, where there are two transformer blocks for two sequence dimensions separately. More details can be found in Appendix A.1. We choose state-of-the-art sequence parallel methods including DeepSpeed-Ulysses (Jacobs et al., 2023), Megatron-SP (Korthikanti et al., 2022), Megatron-LM (Shoeybi et al., 2019) and RingAttention (Liu et al., 2023a) as baselines.

### 4.1. Communication Analysis

The primary advantage of DSP lies in its ability to minimize communication costs and enable scalable communication operations. DSP exploits the inherent characteristics of multi-dimensional transformers to eliminate unnecessary communication, compared with embedded approaches such as Megatron-LM (Shoeybi et al., 2019), Megatron-SP (Korthikanti et al., 2022), RingAttention (Liu et al., 2023a) and DeepSpeed-Ulysses (Jacobs et al., 2023). Consider an activation size of $M$ and a sequence parallel size of $N$. In a 2D-Transformer, there is one transformer block for each sequence dimension per layer, resulting in two transformer

*Table 3.* Comparison of DSP with other sequence parallelism methods for 2D-Transformer architectures. $M$ denotes the activation size, and $N$ represents the number of devices. Communication volume refers to the per-layer (two 1D blocks) volume per device.

| Method | Communication Volume | Activation Memory | Parameter Memory | Ease of Use |
|---|---|---|---|---|
| Ring Attention | $2M$ 👎 | 👍 | 👍 | 👎👎 |
| Megatron-LM | $8M$ 👎👎 | 👎 | 👍 | 👎 |
| Megatron-SP | $8M$ 👎👎 | 👎 | 👍 | 👎 |
| DeepSpeed-Ulysses | $4M/N$ 👍 | 👍 | 👍 | 👍 |
| DSP (ours) | $2M/N$ 👍👍 | 👍👍 | 👍 | 👍👍 |

blocks per layer (two 1D blocks). More details are demonstrated in Appendix A.3.

**Megatron-LM & Megaton-SP** both need to gather and scatter the whole sequence. Megatron-LM utilizes 2 all-reduce per block for attention and mlp, while Megatron-SP employs 2 all-gather and 2 reduce-scatter operations per block. They lead to a total per-device communication volume of $8M$ for both methods.

**Ring-Attention** needs to communicate the entire key and value in the temporal block only, resulting in a total per-device communication volume of $2M$.

**DeepSpeed-Ulysses** incurs 4 all-to-all in temporal block for the query, key, value, and output of attention, resulting in a total per-device communication volume of $4M/N$.

**DSP** mitigates communication cost by employing only two all-to-all operations in total two blocks per layer. As shown in Table 4.1, it reduces the communication volume to $2M/N$, significantly outperforming other sequence parallelism techniques. Notably, with merely two all-to-all operations, DSP exhibits efficient scalability even in super-large clusters for training and inference on extremely long sequences because the communication volume decreases as the number of nodes increases, rendering DSP an exceptional choice for large-scale distributed training and inference tasks involving extreme long sequences.

### 4.2. Memory Analysis

Regarding activation memory, since we shard every tensor in the transformer, we are theoretically able to achieve the minimum activation cost, similar to DeepSpeed-Ulysses and Ring Attention. In practice, however, our approach requires less reshape and communication overhead, allowing us to further reduce intermediate activation memory compared to other methods. Megatron-SP and Megatron-LM, on the other hand, needs to hold some entire sequences, resulting in higher memory requirements.

As for parameter memory, as discussed in Section 3.4, we utilize the ZeRO technique (Rajbhandari et al., 2019) to

shard all parameters across different devices to ensure low parameter memory footprint.

## 5. Experiments

Experiments are conducted on 128 NVIDIA H100 GPUs, interconnected via NVLink within nodes and InfiniBand across nodes. Our methods and implementations are not dependent on specific hardware architectures and can generalize to other devices, particularly those with less efficient interconnects. We follow the same baseline and settings as discussed in Section 4, utilizing 720M and 3B size Transformer-2D models in our experiments. Despite the existence of various 2D-Transformer variants, their architectures are fundamentally similar. We select one model similar to OpenSora (Zangwei Zheng, 2024). The code is implemented using PyTorch (Paszke et al., 2019).

In the following evaluations, we focus on addressing the following questions: 1) How is DSP's end-to-end performance compared with other SOTA sequence parallelism? 2) How is DSP's scaling ability when scale to many GPUs? 3) What is DSP's memory consumption like in practice?

### 5.1. End-to-End Performance

In this section, we compare the end-to-end performance of different sequence parallelism methods on 128 NVIDIA H100 GPUs. We use a combination of sequence parallelism and data parallelism, with the sequence parallelism set to the minimum size for each method. We evaluate across different sequence lengths ranging from 0.5 million to 4 million tokens, which are common usages for video generation. Details can be found in Appendix A.2.2. As shown in Figure 5, DSP is able to outperform DeepSpeed-Ulysses by 32% to 75%, and other methods by up to 10x due to its communication efficiency. As the sequence length becomes longer and the sequence parallel size increases, as DSP's communication volume decreases as the device number increases, our performance's advantage over the baselines becomes even more pronounced. When scaling from 0.5M to 4M tokens, our FLOPS drops by at most 23%, while other

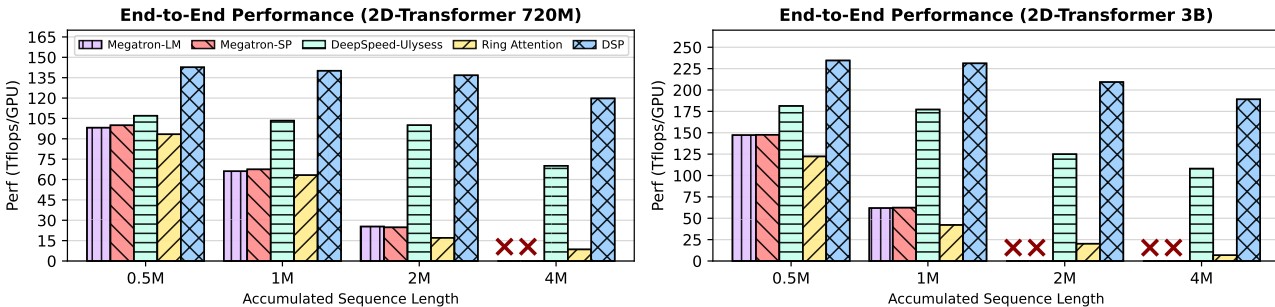

Figure 5. End-to-end performance comparison of different sequence parallel methods combined with data parallel on 128 H100 GPUs. The sequence parallel size is set to minimum for each method.

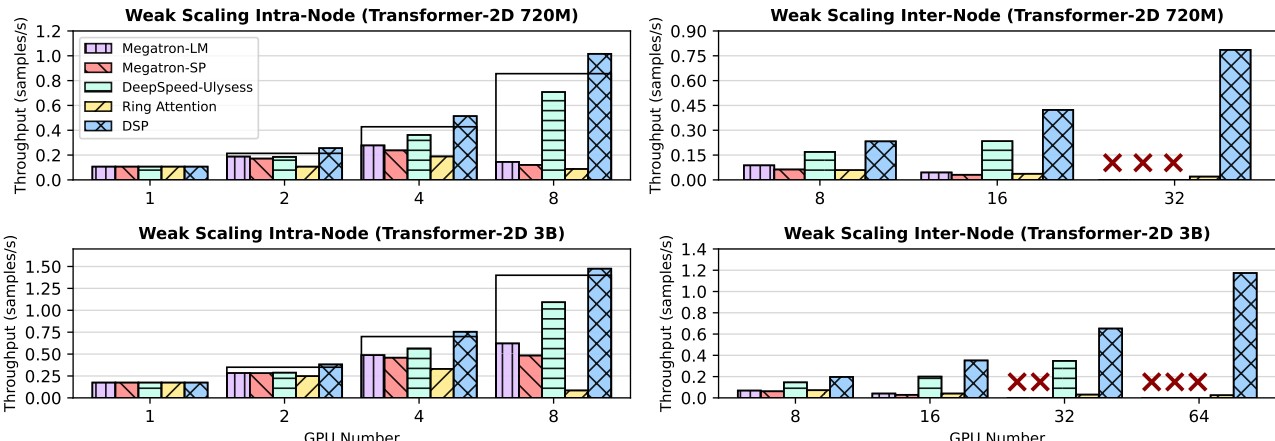

Figure 6. Weak scaling ability evaluation of different methods with sequence parallelism only. "×" denotes out of memory or head. Black boxes represent linear scaling.

methods experience at least a 40% drop.

## 5.2. Scaling Ability

This section evaluates the scaling ability of DSP from two perspectives: weak scaling and strong scaling. Weak scaling refers to scenarios where the computational workload per device remains constant while incrementally increasing the number of devices. This setup is analogous to the training stage, where the goal is to scale longer sequences over more GPUs. Strong scaling, on the other hand, is more challenging as it requires keeping the total computational workload constant while incrementally increasing the number of devices. In this case, the computation becomes more sparse on each device. Strong scaling is often employed when the objective is to infer an input sequence rapidly across many GPUs for low-latency applications. The experiments are divided into intra-node and inter-node evaluations due to the different interconnection conditions. Intra-node experiments leverage NVLink interconnect for communication, while inter-node experiments utilize a combination of NVLink and InfiniBand interconnect. More details can be found in Appendix A.2.3.

**Weak Scaling.** In the weak scaling experiments, to maintain a consistent computational workload for each GPU, the batch size is linearly increased proportional to the number of GPUs, while the sequence length is fixed. As shown in Figures 6, DSP significantly outperforms other methods by more than 80.7%. Moreover, DSP can scale up to 64 GPUs without being limited by the number of attention heads, unlike DeepSpeed-Ulysess and Megatron-SP. Despite scaling to 64 GPUs, DSP maintains an almost linear throughput increase, with only a 15% performance loss from 8 GPUs to 64 GPUs. Additionally, DSP can achieve super-linear scaling for intra-node due to efficient communication.

**Strong Scaling.** In the strong scaling experiments, both batch size and sequence length are fixed. As shown in Figure 7, DSP can maintain linear scalability when scaling up to 8 GPUs for 720M model and 4 GPUs for 3B model, which covers most practical scenarios. To evaluate the extreme performance capabilities of DSP, we further scale up to 64 GPUs with very little workload per device. Although there is an inevitable performance drop, DSP's throughput remains significantly better than the baselines. As shown in Figure 8, our work can significantly reduce inference

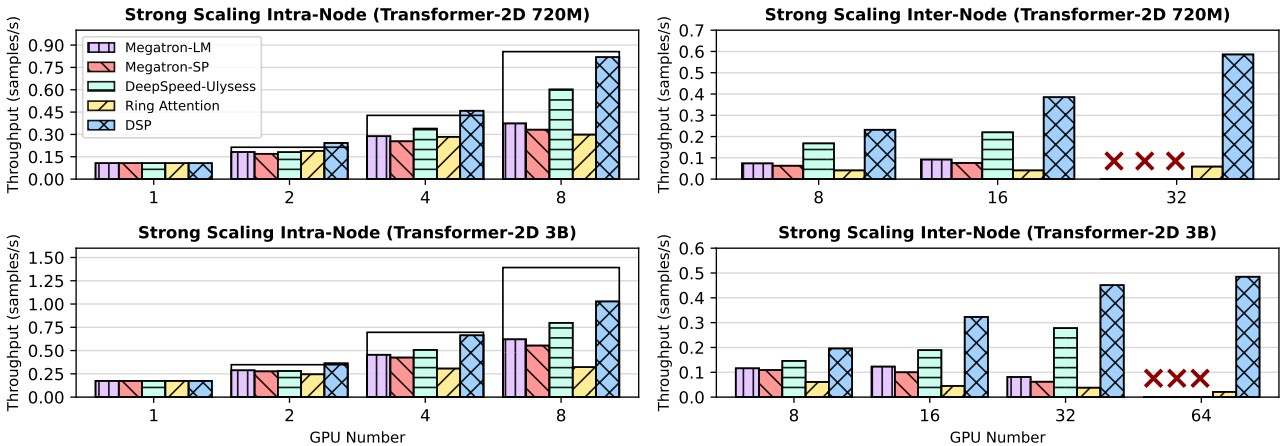

Figure 7. Strong scaling ability evaluation of different methods with sequence parallelism only. "×" denotes out of memory or head. Black boxes represent linear scaling.

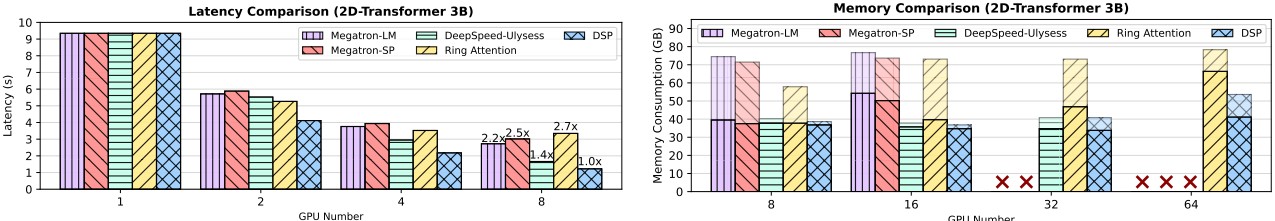

Figure 8. Inference latency comparison of different sequence parallelism methods.

Figure 9. Memory comparison of different sequence parallelism methods.

latency compared with baselines with the same workload.

## 5.3. Memory Consumption

Figure 9 demonstrates the memory consumption comparison of different baselines in the weak scaling setting. The semi-transparent bar represents the cached memory, while the solid bar represents the allocated memory. The total memory usage is the sum of them. Our approach exhibits the lowest memory usage, scaling efficiently for longer sequences. Furthermore, DSP's memory usage is compact without excessive cache memory bloat, unlike Ring-Attention, Megatron-LM and Megatron-SP.

## 6. Conclusion

In this work, we introduce Dynamic Sequence Parallelism (DSP), a novel sequence parallel abstraction for effectively scaling multi-dimensional transformers to long sequences. Unlike current embedded sequence parallel methods that only shard on single sequence dimension and are tailored to specific patterns, DSP offers a general and elegant solution by dynamically switching the parallel dimension during computation, decoupled from the computation module.

The key advantages of DSP are: 1) substantially reduced

communication costs, 2) adaptability across modules without specialized modifications, and 3) remarkable ease of implementation enabled by a simple high-level API. Our experiments demonstrated DSP's superiority, achieving from 32.2% to 10× higher end-to-end throughput and at least 75% lower communication volume compared to state-of-the-art methods. Its elegance and ease of use make it a promising solution for efficient sequence parallelism across a wide range of applications.

**Limitations.** One limitation of this work is that DSP is specifically designed for multi-dimensional transformers and may not adapt well to single-dimensional ones like language models. Additionally, while there are global operations that involve all sequence dimensions, which are rare in transformer, DSP may not be of optimal efficiency.

**Future works.** In the future, DSP could expand its scope beyond transformer architectures to architectures including convolution, recurrent, and graph neural networks to utilize its potential across various tasks. Furthermore, automated optimization techniques could enable DSP to dynamically and autonomously determine the most effective switching strategy based on network analysis, thereby optimizing overall system efficiency and efficacy.

## Impact Statement

This paper presents work whose goal is to advance the field of Machine Learning. There are many potential societal consequences of our work, none which we feel must be specifically highlighted here.

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

# DSP: Dynamic Sequence Parallelism for Multi-Dimensional Transformers

## Appendix

We organize our appendix as follows:

## A. Appendix

### A.1. Model Details

In the theoretical analyses and evaluation section, we use a Transformer-2D model as our base model, similar to OpenSora (Zangwei Zheng, 2024). However, it is not exactly OpenSora; we have removed its specific cross-attention module to ensure that the performance can be generalized to other models. Therefore, in each layer, there are only two transformer blocks that process two sequence dimensions separately, as shown in Figure 10. Specifically, the two dimensions are temporal $t$ and spatial $s$ for a sequence. Each dimension is processed by a corresponding transformer block, which is a common strategy in many applications.

### A.2. Experiment Settings

#### A.2.1. MODEL SIZE

*Table 4.* End-to-end performance parallel settings. Tuple for methods denotes (sequence_parallel_size, data_parallel_size).

| Model Size | Sequence Length | Temporal Sequence | Spatial Sequence | DeepSpeed Ulysses | Megatron SP | Ring Attention | DSP |
|---|---|---|---|---|---|---|---|
| 720M | 0.5M | 128 | 4096 | (2, 64) | (2,64) | (2, 46) | (2, 64) |
|  | 1M | 256 | 4096 | (4, 32) | (4,32) | (4, 32) | (4, 32) |
|  | 2M | 512 | 4096 | (8, 16) | (16,8) | (8, 16) | (8, 16) |
|  | 4M | 1024 | 4096 | (16, 8) | / | (16, 8) | (16, 8) |
| 3B | 0.5M | 128 | 4096 | (4, 32) | (4, 32) | (4, 32) | (4, 32) |
|  | 1M | 256 | 4096 | (8, 16) | (16,8) | (8, 16) | (8, 16) |
|  | 2M | 512 | 4096 | (16, 8) | / | (16, 8) | (16, 8) |
|  | 4M | 1024 | 4096 | (32, 4) | / | (32, 4) | (32, 4) |

*Table 5.* Model settings of 720M and 3B 2D-Transformer.

| Model Name | Layers | Hidden States | Attention Heads | Patch Size |
|---|---|---|---|---|
| 720M | 28 | 1152 | 16 | (1, 2, 2) |
| 3B | 36 | 2038 | 32 | (1, 2, 2) |

In the experiments, we use 720M and 3B size for 2D-Transformer. There specific model settings are shown in Table 5.

### A.2.2. End-to-end Performance

Here is the polished text in a more formal format without using bullet points: In end-to-end performance experiments, 128 GPUs were utilized for all methods. For each method, the minimum sequence parallel size that would not result in out-of-memory errors was employed to reduce communication overhead, with data parallelism employed for the remaining size. ZeRO-2 was used for all methods except Megatron-SP. The specific parallel size is detailed in Table 4.

The accumulated sequence length ranged from 0.5M to 4M, which appears significantly larger than typical text lengths. However, such lengths are common for multi-dimensional tasks. In this case, we followed the workload of video generation. The spatial sequence, representing video resolution, was fixed at 1024x1024. After applying the Variational Autoencoder (VAE) and Patch Embedding, the final length for the spatial sequence was 4096. The temporal sequence, representing video length, scales linearly in the test.

### A.2.3. Scaling Ability

*Table 6.* Strong scaling experiment settings.

| Model Size | Type | Batch Size | Temporal | Spatial |
|---|---|---|---|---|
| 720M | Intra-Node | 1 | 64 | 4096 |
| | Inter-Node | 1 | 256 | 4096 |
| 3B | Intra-Node | 1 | 16 | 4096 |
| | Inter-Node | 1 | 128 | 4096 |

*Table 7.* Weak scaling experiment settings.

| Model Size | Type | GPU Number | Batch Size | Temporal | Spatial |
|---|---|---|---|---|---|
| 720M | Intra-Node | 1 | 1 | 64 | 4096 |
| | | 2 | 2 | 64 | 4096 |
| | | 4 | 4 | 64 | 4096 |
| | | 8 | 8 | 64 | 4096 |
| | Inter-Node | 8 | 1 | 256 | 4096 |
| | | 16 | 2 | 256 | 4096 |
| | | 32 | 4 | 256 | 4096 |
| 3B | Intra-Node | 1 | 1 | 16 | 4096 |
| | | 2 | 2 | 16 | 4096 |
| | | 4 | 4 | 16 | 4096 |
| | | 8 | 8 | 16 | 4096 |
| | Inter-Node | 8 | 1 | 128 | 4096 |
| | | 16 | 2 | 128 | 4096 |
| | | 32 | 4 | 128 | 4096 |
| | | 64 | 8 | 128 | 4096 |

In weak scaling experiments, as shown in Figure 7 we fix the sequence length and linearly increase the batch size, ensuring that the workload on each device remains constant as the number of devices scales. In strong scaling experiments, as shown in Figure 6, we fix both the sequence length and batch size, keeping the total computation constant. For each experiment, we set the sequence length to the maximum for the least GPU case to fully utilize the computational resources. Specifically, we use the same spatial sequence length and adjust the temporal sequence length to its maximum for each test and sequence parallel size is set to GPU number.

### A.3. Parallelism Implementation

In Figure 10, we demonstrate the detailed implementation of different sequence parallel methods on 2D-Transformer. The implementation of DeepSpeed-Ulysses (Rasley et al., 2020), Megatron-SP (Korthikanti et al., 2022) and Megatron-LM

(Shoeybi et al., 2019) are adopted based on their official implementation. For Ring-Attention (Liu et al., 2023a), we adopt an unofficial implementation and adapt it to the 2D-Transformer.

| Communication Type | Communication Volume |
|---|---|
| all-reduce | 2M |
| all-gather | M |
| reduce-scatter | M |
| all-to-all | M/N |

*Table 8.* Per-device communication volume for each operation.

| Method | Communication Type | Communication Times Per Layer (Two 1D Blocks) | Time (ms) |
|---|---|---|---|
| DSP | all-to-all | 2 | 914 |
| DS-Ulysses | all-to-all | 4 | 5828 |
| Megatron-LM | all-reduce | 4 | 8412 |
| Megatron-SP | all-gather + reduce-scatter | 4 | 51908 |

*Table 9.* Comparison of communication methods and performance with a 8M length of sequence.

Megatron-SP employs four resource-intensive collective communication operations per transformer block. Specifically, it initiates an all-gather operation to aggregate the entire input $x$, succeeded by reduce-scatter operations at the output for both attention and MLP modules, culminating in a total communication volume of $8M$ for one layer (two 1D blocks). Note that the communication volume is calculated per device.

Similarly, Megatron-LM employs 2 all-reduce per transformer block, culminating 4 all-reduces, in a total communication volume of $8M$.

DeepSpeed-Ulysses adopts the more efficient all-to-all approach. It leverages all-to-all for query, key, value to transform their shard dimension before attention, and a all-to-all for output after attention. And it only need to communicate in temporal transformer block. Consequently, the communication volume transmitted per device for an AlltoAll communication of size $M$ across $N$ GPUs is $4M/N$.

Ring-Attention is not shown in the figure because it does not require resharding. We implement sequence communication in the temporal transformer as the time axis is split. In the attention module, it needs to pass the key and value to all other devices, resulting in a total communication volume of $2M$.

DSP applies dynamic switching between stages to switch the parallel dimension, which involves two AlltoAll operations, totaling $2M/N$ communication.

As shown in Table 8 and 9, we demonstrate the communication volume for each communication method, and their latency for a sequence of 8M length.

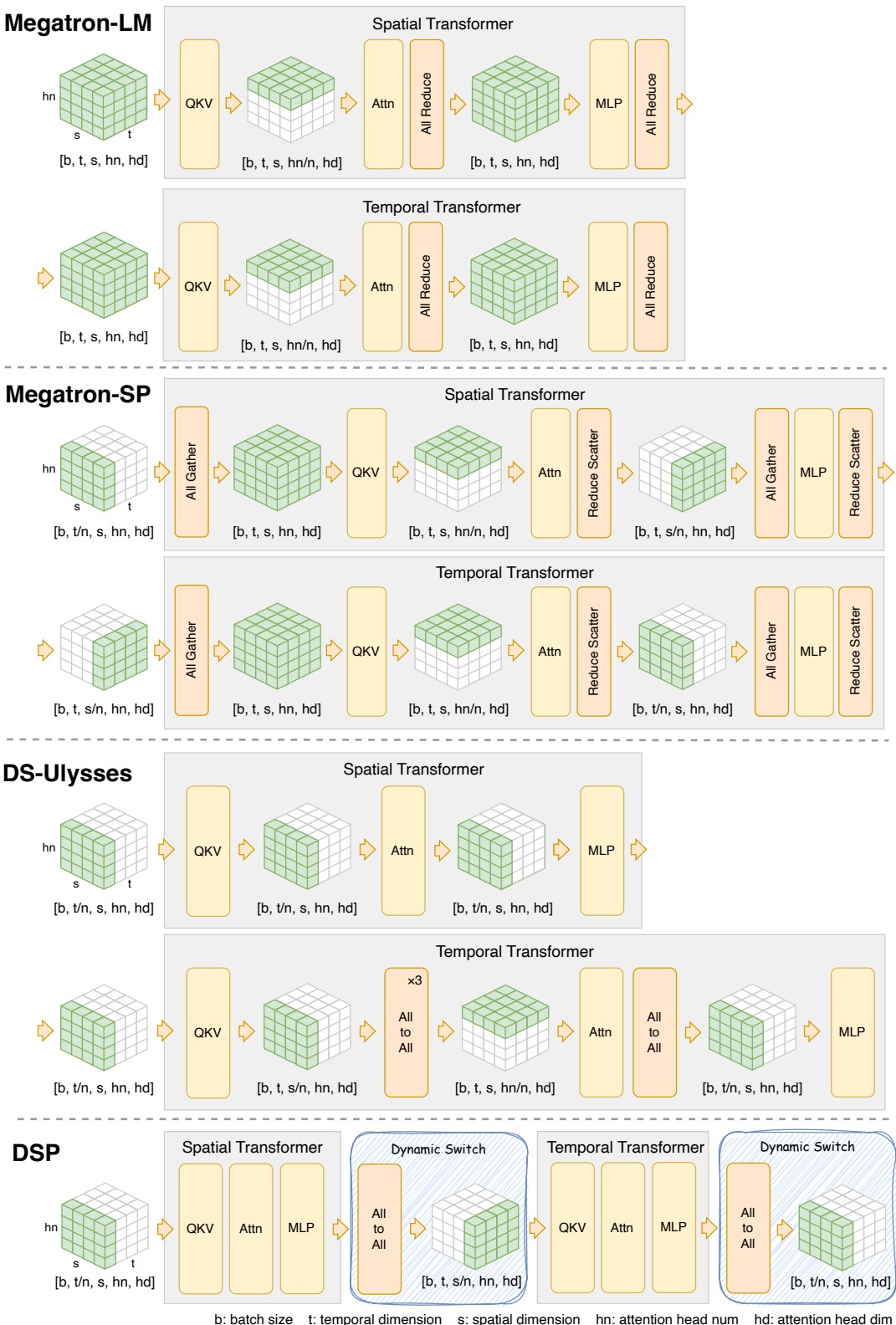

*Figure 10.* Overview of different sequence parallelism methods for 2D-Transformer.

