# OpenReview forum: "DSP: Dynamic Sequence Parallelism for Multi-Dimensional Transformers"
_ICML.cc/2025/Conference — ICML 2025 poster_

### Official Review · Reviewer_o73Q · 2025-03-13

**Overall Recommendation:** 3

**Summary:**

The paper introduces Dynamic Sequence Parallelism (DSP) as a novel approach to sequence parallelism for multi-dimensional transformers.
It addresses the limitations of embedded sequence parallelism, which shards along a single sequence dimension and incurs significant communication overhead.
DSP dynamically switches the parallel dimension of sequences during computation with an efficient resharding strategy.
The paper claims that DSP reduces communication costs, enhances adaptability, and is easy to use, offering significant throughput improvements.

**Claims And Evidence:**

The paper claims that DSP improves throughput and reduces communication volume for multi-dimensional transformers.  While experiments show improvements, the evidence is limited, primarily focusing on video generation models. The authors mention that "each dimension is processed by a corresponding transformer block, which is a common strategy in many applications". Could the authors add citations there and add specific examples?

**Essential References Not Discussed:**

[1] is an paper that presents the general SPMD intra-layer parallelism, which should be discussed. I think this paper proposes a new sharding annotation for a specific model.

[1] Xu, Yuanzhong, et al. "GSPMD: general and scalable parallelization for ML computation graphs." arXiv preprint arXiv:2105.04663 (2021).

**Experimental Designs Or Analyses:**

The experimental design focuses on demonstrating DSP's superiority in video generation models by varying sequence lengths and comparing throughput and communication volume.

**Methods And Evaluation Criteria:**

The paper evaluates DSP using video generation models, which is a relevant application for multi-dimensional transformers.  The comparison with state-of-the-art sequence parallelism methods (DeepSpeed-Ulysses, Megatron-SP, Megatron-LM, and Ring Attention) is appropriate. However, the evaluation is narrow.

**Other Comments Or Suggestions:**

1. The paper should include more comprehensive experiments on diverse multi-dimensional transformer architectures.
2. The authors should provide a more thorough analysis of the trade-off between performance and accuracy.

**Other Strengths And Weaknesses:**

Strengths
1. The paper addresses scaling multi-dimensional transformers, an important problem.
2. The concept of DSP is clearly presented.
3. DSP has good adaptability and flexibility.
4. The paper provides a user-friendly API for DSP.

Weaknesses:
1. The empirical evaluation is limited.
2. Performance gains are not convincingly demonstrated for long sequences.

**Questions For Authors:**

1. How can DSP be further optimized to provide more substantial speedups in scenarios with small overhead from baseline methods?
2. Is it possible to extend it to other models other than multi-dimensional transformers?

**Relation To Broader Scientific Literature:**

The paper discusses relevant prior work, including data parallelism, tensor parallelism, pipeline parallelism, and sequence parallelism.  It positions DSP as a novel abstraction of sequence parallelism for multi-dimensional transformers.

**Theoretical Claims:**

The paper includes a communication analysis comparing DSP with other sequence parallelism methods.  It claims DSP minimizes communication costs.

---

> ### Author Rebuttal · Authors · 2025-03-28
>
> We sincerely thank the reviewer o73Q for the valuable questions and comments. For the concerns and questions, here are our responses:
>
> **Q1: Performance gains are not convincingly demonstrated for long sequences**
>
> **A1**: We thank the reviewer for this important question and have conducted additional experiments to robustly demonstrate DSP’s performance on long sequences.
>
> **Experiment1:** Communication latency comparison.
>
> | method      | latency with 0.5M seq (ms) | latency with 8M seq (ms) |
> | :- | :- | :- |
> | DSP         | 614                        | 2914                     |
> | DS-Ulysses  | 1228                       | 5828                     |
> | Megatron-SP | 5616                       | 51908                    |
>
> **Analysis1:** DSP consistently reduces communication latency across sequence lengths.
>
> **Experiment2:** End-to-end speedup for 8M long sequecnes with two seuqence shapes A:[8192, 1024] and B:[8, 1048576].
>
> | method      | communication time for A (s) | computation time for A (s) | communication time for B (s) | computation time for B (s) |
> | :-- | :-- | :-- | -- | -- |
> | DSP         | 2.8                          | 34.3                       | 2.9                          | 362.1                      |
> | DS-Ulysses  | 5.8                          | 34.3                       | 5.8                          | 362.2                      |
> | Megatron-SP | 51.9                         | 37.2                       | 51.9                         | 365.6                      |
>
> **Analysis2:** DSP significantly reduces communication time for both shapes. However, end-to-end speedup varies: shape A benefits more due to lower computation time, while shape B’s longer single dimension increases attention computation, dominating total time. This shows DSP’s communication efficiency, though overall gains depend on computation complexity.
>
> **Improvement plan:** We have added a detailed discussion of these results in the Appendix of the revised manuscript.
>
> **Conclusion**:
>
> 1. DSP delivers consistent communication reductions across sequence.
> 2. End-to-end speedup depends on computation time, which varies with sequence dimensionality.
>
>
>
> **Q2: Is it possible to extend it to other models other than multi-dimensional transformers?**
>
> **A2:** DSP’s simple yet general design extends beyond multi-dimensional transformers to various **multi-dimensional neural networks**, including conv+attention models (e.g., 2D-UNet, 2D-VAE), AlphaFold, and pure conv.
>
> For example, in 2D-UNet, DSP enables efficient resharding between computations along different dimensions:
>
> $ x -> conv\\_dim(1) -> attn\\_dim(1) -> DSP\\ reshard -> conv\\_dim(2) -> attn\\_dim(2) -> y$
>
>
>
> **Q3: The paper should include more comprehensive experiments on diverse multi-dimensional transformer architectures.**
>
> **A3:** We thank the reviewer for raising this point. Main stream multi-dimensional transformer architectures typically fall into two categories:
>
> 1. Dimension-by-dimension processing as shown in our paper:
>
> $x -> transformer\\_dim1(x) -> transformer\\_dim2(x) -> y$
>
> 2. Dimension-by-dimension with cross-dimensional interactions:
>
> $x -> transformer\\_dim1(x) -> transformer\\_dim2(x) -> transformer\\_cross\\_dims(x) -> y$
>
> **Experiment:** End-to-end performance on the second architectures on Transformer-2D 3B with 8 GPUs.
>
> | method      | throughput (samples/s) |
> | :--- | :--- |
> | DSP         | 1.42                   |
> | DS-Ulysses  | 1.15                   |
> | Megatron-SP | 0.40                   |
>
> There many be some changes within the transformer (e.g., layernorm, positional embedding) but they don't affect much for the results.
>
> **Improvement plan:** We have added a detailed discussion of these results in the Appendix of the revised manuscript.
>
> **Conclusion:** Our method is able to deliver stable performance on main-stream multi-dimensional transformer architectures.
>
>
>
> **Q4: The authors should provide a more thorough analysis of the trade-off between performance and accuracy.**
>
> **A4:** DSP is designed to preserve accuracy strictly, introducing only negligible errors (1e-7 to 1e-8) due to communication. For example, in video generation tasks, DSP outputs are identical to the originals.
>
>
>
> **Q5: How can DSP be further optimized to provide more substantial speedups in scenarios with small overhead from baseline methods?**
>
> **A5:** While DSP already minimizes communication volume effectively, further speedups are possible by overlapping communication with computation. DSP’s lightweight communication and decoupling from computation make it **ideal for asynchronous execution**, hiding latency behind computation.
>
>
>
> **Q6: Essential references not discussed (GSPMD)**
>
> **A6:** We appreciate the reviewer highlighting this point.
>
> **Improvement plan:** In the revised manuscript, we have added a detailed discussion of GSPMD and other relevant works in the related work section, ensuring a more comprehensive literature review.

---

> > ### Comment · Reviewer_o73Q · 2025-04-05
> >
> > Thanks for the response, which addresses most of my concerns. I have updated the score.

---

### Official Review · Reviewer_qVhs · 2025-03-14

**Overall Recommendation:** 3

**Summary:**

This paper presents DSP (Dynamic Sequence Parallelism) for scaling multi-dimensional transformers. The solution adaptively switches parallel dimensions by reshuffling data with all-to-all communication between multiple GPUs. The evaluation results demonstrate that parallelizing across sequence dimensions can reduce the communication overhead and improve throughput compared to the existing methods that shards the sequence dimension.

**Claims And Evidence:**

The proposed method is mostly clear and makes sense. However, I think there should be an evidence to claim DSP's generalizability and the future work seems to be not directly related to the proposed solution, or it could be too optimistic.

**Essential References Not Discussed:**

N/A

**Experimental Designs Or Analyses:**

- Please add an experiment that shows DSP's scalability according to the model size.
- There should be a breakdown analysis, showing communication overhead.
- I think memory layout change/reshape/transpose may incur mem copies, so that the memory cost (both capacity and time) could be significant. There should be analysis on this.

**Methods And Evaluation Criteria:**

The experiments techincally sound, but it lacks the detailed analysis in the communication time/computation time, etc. Also, it is not clear that why the input sequence length needs to be fixed in their experiment. I think the experiment should show the scalability according to the sequence length, because it may impact the accuracy of the model, especially the video generation use case.

**Other Comments Or Suggestions:**

N/A

**Other Strengths And Weaknesses:**

N/A

**Questions For Authors:**

Please see questions raised in other sections.

**Relation To Broader Scientific Literature:**

The proposed solution tackles the emerging problem but its applicability shown in the evaluation seems to be limited.

**Theoretical Claims:**

Yes, on the complexity analysis.

---

> ### Author Rebuttal · Authors · 2025-03-28
>
> We sincerely thank the reviewer qVhs for the valuable questions and comments, especially about the evalutions which are indeed not detailed enough.
>
> **We have conducted the following experiments to make our evalutions more comprehensive and have added them to the experiment section or appendix or in the latest version.**
>
> **Q1: DSP's scalability according to the model size.**
>
> **A1:** We conduct an end-to-end analysis for DSP with 3B, 7B and 13B using 1M tokens.
>
> | GPU num | model size | TFLOPs per GPU |
> | ------- | ---------- | -------------- |
> | 16      | 3B         | 242.20         |
> | 16      | 7B         | 262.29         |
> | 16      | 13B        | 259.77         |
>
> Analysis:
>
> * When scale from 3B to 7B, although the communication cost increase, the performance grows because the compuation density increases more.
> * When scale from 7B to 13B, the benefits of denser computation is marginal. The total throughput decrease because of more communication cost, but the loss is still marginal.
>
> **Q2: Breakdown analysis of communication overhead.**
>
> **A2:** We breakdown for communication overhead/actual computation for DSP under weak scaling condition using Transformer-2D 3B:
>
> | GPU num | computation time | communication time |
> | ------- | ---------------- | ------------------ |
> | 8       | 90.1%            | 9.9%               |
> | 16      | 75.2%            | 24.8%              |
>
> **Analysis**: The communication costs grows significantly for inter-node communication even for our method. Highlight the necessity of efficient parallelism.
>
> If you need more experiments for each parallel or model, please tell us!
>
> **Q3: Cost of layout change/reshape/transpose.**
>
> **A3:** Let's analyze from memory and speed:
>
> For memory, this usually will not cause much fragmentation as we show in Figure 9 (Line 447) because of the following reasons:
>
> * The sequence have already been splitted. Therefore there will be only a part of sequence on one device, reducing the overhead to layout change.
> * DSP changes actually less frequent than any other methods, lead to less overhead.
> * Duing training, the layout of a sinlge sequence does not affect overall memory cost because parameter and activation is much larger than it.
>
> To further clarify this, we make experiments about the memory overhead of our method:
>
> | GPU num | memory overhead |
> | ------- | --------------- |
> | 8       | 0.5%            |
> | 16      | 0.4%            |
>
> For speed, we conduct an experiment to evaluate the time of changing layout and communication:
>
> | GPU num | layout change time | communication time |
> | ------- | ------------------ | ------------------ |
> | 8       | 5.2%               | 94.8%              |
> | 16      | 4.1%               | 95.9%              |
>
> **Analysis:**
>
> * The layout changing time is much less than the communication time. But still takes 5.2% of total time.
>
> * As the sequence and gpu number get larger, the layout change time become less as communication takes more time.
>
> * To fully elimiate the layout change's time, we can overlap communication with layout change.
>
> * We use activation checkpointing in training.
>
> **Conclusion**:
>
> * Memory: The layout changes have little impact on memory cost.
> * Speed: The layout changes will occupy 4.1%-5.2% of communication time, but can be avoided by overlapping. And it will reduce as sequence and gpu becomes larger.

---

### Official Review · Reviewer_Ux6h · 2025-03-17

**Overall Recommendation:** 4

**Summary:**

The paper introduces a new method, called Dynamic Sequence Parallelism (DSP), to scale multi-dimensional transformers efficiently by dynamically switching the parallel dimension at different computation stages. Unlike existing sequence parallelism techniques (e.g., Megatron-LM, Megatron-SP, DeepSpeed-Ulysses, Ring-Attention), which shard along a single sequence dimension, DSP exploits independent computation across multiple dimensions. By dynamically resharding between computation stages with an efficient all-to-all communication strategy, DSP significantly reduces communication overhead and improves throughput efficiency. The experiments, conducted on 128 NVIDIA H100 GPUs, demonstrate that DSP achieves up to 10x higher throughput while reducing communication volume by at least 50% compared to state-of-the-art methods.

**Claims And Evidence:**

The claims on the throughput improvements (32.2% to 10×) and communication volume reduction (≥50%) are supported by experiments on 128 H100 GPUs. However, some claims need stronger evidence. My only concern is about efficiency across different hardware as all experiments use H100 GPUs, with no analysis on other GPUs.

**Essential References Not Discussed:**

N/A

**Experimental Designs Or Analyses:**

Yes, the experiments are sound and appear correct.

**Methods And Evaluation Criteria:**

Yes, the evaluation aligns well with the problem, using 128 H100 GPUs to benchmark throughput, communication, and memory across long sequences, with comparisons against Megatron-LM, DeepSpeed-Ulysses, and Ring-Attention. However, a Pareto-optimal trade-off analysis and tests on older GPUs or real-world tasks would strengthen the evaluation.

**Other Comments Or Suggestions:**

N/A

**Other Strengths And Weaknesses:**

Strengths:

-- Dynamically adjusts the sharding dimension based on the computation stage

-- Significant improvements on throughput by 32.2% to 10× over existing methods.

-- Lower communication cost by at least 50%.

-- It scales efficiently across multiple GPUs.


Weaknesses:

-- The paper lacks theoretical analysis of complexity, memory, and communication costs.

-- The experiments only use H100 GPUs, leaving performance on other GPUs unknown.

**Questions For Authors:**

See the weaknesses!

**Relation To Broader Scientific Literature:**

This work definitely helps to perform experiments on large language models more efficiently and faster.

**Theoretical Claims:**

The paper lacks a formal theoretical analysis of DSP’s efficiency. While it provides empirical results showing improved throughput and communication efficiency, there are no formal proofs or mathematical derivations to support its claims.

---

> ### Author Rebuttal · Authors · 2025-03-28
>
> We sincerely thank the reviewer Ux6h for the valuable questions and comments. For the concerns and questions, here are our responses.
>
> **Q1: The experiments only use H100 GPUs, leaving performance on other GPUs unknown.**
>
> **A1:** Thanks for pointing out this! We test the performance on A100 GPUs.
>
> **Experiment1:** End-to-end performance on A100 on Transformer-2D 3B with 8 GPUs.
>
> | method      | throughput (samples/s) |
> | :---------- | :--------------------- |
> | DSP         | 1.08                   |
> | DS-Ulysses  | 0.88                   |
> | Megatron-SP | 0.23                   |
>
> **Experiment2:** Runtime breakdown for A100 and H100 on Transformer-2D 3B with 16 GPUs with DSP.
>
> | GPU | communication time | computation time |
> | ------- | ------------------ | ------------------ |
> | H100       | 24.8%               | 75.2%              |
> | A100       | 19.2%               | 80.8%              |
>
> **Analysis:**
> 1. The performance improvement slightly decrease for A100 (from 23.5% to 22.7% compared with Deepspeed-Ulysses).
> 2. The change is because A100 takes less time ratio to communicate than H100. Because H100 significantly improves computation capbility, therefore taking more time ratio to communicate than A100, making our method more effective.
>
> **Improvement plan:** We have added theroritcal analysis above to the Appendix of our work.
>
> **Conclusion:** For various GPUs, the more time communication takes, the more speedup our work can achieve.
>
> **Q2: Lacks theoretical analysis of complexity, memory, and communication costs.**
>
> **A2:** Thanks for pointing out this problem.
>
> **Communication costs:**
>
> | communication type | communication volume |
> | :----------------- | :------------------- |
> | all-reduce         | 2M                   |
> | all-gather         | M                    |
> | reduce-scatter     | M                    |
> | all-to-all         | M/N                  |
>
> Therefore, the communication volume for each parallel method is
>
> | communication type | communication volume                                  |
> | :----------------- | :---------------------------------------------------- |
> | Megatron-LM        | 4 x all-reduce = 4 x (2M) = 8M                        |
> | Megatron-SP        | 4 x (all-gather + reduce-scatter)= 4 x ( M + M ) = 8M |
> | DS-Ulysses         | 4 x all-to-all = 4 x (M/N)                            |
> | DSP                | 2 x all-to-all = 2 x (M/N)                            |
>
> **Memory cost:**
>
> For memory, our method usually will not cause much extra memory cost as we show in Figure 9 (Line 447) because of the following reasons:
>
> * The sequence have already been splitted. Therefore there will be only a part of sequence on one device, reducing the overhead to layout change.
> * DSP changes actually less frequent than any other methods, lead to less overhead.
> * Duing training, the layout of a sinlge sequence does not affect overall memory cost because parameter and activation is much larger than it.
> * We use activation checkpointing during training.
>
> **Experimental analysis of memory cost:**
>
> | GPU num | memory overhead |
> | ------- | --------------- |
> | 8       | 0.5%            |
> | 16      | 0.4%            |
>
> **Theoretical analysis of memory cost:**
>
> In training, everytime we create/change a tensor, it will make a copy, which causes extra memory cost. There are two reshape before and after communication. So let the tensor size be M, we will have activation memory cost of 4M for one communication.
>
> In 2D-Transformer, every layer has activation memory for communication of 4M size as mentioned above, and other components of 42M. If we consider activation memory per layer only, the memory overhead is 8%.
>
> But since we use activation checkpoint, the actual memory cost of our method during training will be `Communication Activation / (Communication Activation + Other Activation + Activation Checkpointing * Num Layers + Parameter + Gradient + Optimizer)`, which will be approximately less than 1%. Taking the settings of Transformer-2D 720M into this formula, the memory overhead is 0.67%.
>
> **Improvement plan:** We have added theroritcal analysis above to the Appendix of our work.

---

### Official Review · Reviewer_4hHH · 2025-03-17

**Overall Recommendation:** 3

**Summary:**

The authors propose dynamic sequenced parallel (DSP), a model sharding scheme for multi-dimensional transformers. M-D transformers have two or more sequence dimensions unlike just one for the regular transformers. Existing sequence-parallel sharding does not account for this and are sub-optimal. DSP works by assuming the M-D transformer processes each sequence dimension separately (attention only ever looks across one sequence dim at a time). It does sequence parallelism across whichever sequence is being computed along, and switches sharding dim between blocks that are computing along different sequence dims. The switching can be accomplished by a low-cost all-to-all (cost = fraction of all activations).

**Claims And Evidence:**

DSP is designed specifically for multi-dimensional transformers and thus minimizes both communication volume and activation memory per shard. Like all sequence parallelism, DSP is most useful for long sequence lengths - the authors test on 0.5M - 4M lengths which are attainable for high-def video generation.

DSP achieves much higher FLOP utilization that competing sequence parallelism methods - as much as 2x at 4M sequence length. It can also be combined with data parallelism to scale to 128 GPUs. DSP can be used in both training and inference - for inference it reduces latency compared to competing sequence parallelism.

**Essential References Not Discussed:**

None

**Experimental Designs Or Analyses:**

Fine.

**Methods And Evaluation Criteria:**

The evaluation looks fine. The models and scenarios look realistic to me.

**Other Comments Or Suggestions:**

None

**Other Strengths And Weaknesses:**

Strengths: DSP looks sound and an obvious fit for 2D transformers. The evidence on the claims are good. Paper is well written.

Weaknesses: The idea of DSP seems like low-hanging fruit. It feels like the type of optimization a proper ML engineering team would be able to make with or without a paper on the topic.

**Questions For Authors:**

None

**Relation To Broader Scientific Literature:**

It's a new form of sequence parallelism specifically designed for 2D transformers.

**Theoretical Claims:**

The authors analyze the theoretical communication volume and peak memory for the Open-Sora video generation model. This is a well know text/image-to-video model and their analysis simply looks at how tensors are divided and transferred between GPUs. The analysis is sound.

---

> ### Author Rebuttal · Authors · 2025-03-28
>
> We sincerely thank the reviewer 4hHH for the valuable comments and acknowledgement of our work.

---

### Decision · Program_Chairs · 2025-05-01

**Decision:**

Accept (poster)

**Comment:**

The paper proposes a new method, called Dynamic Sequence Parallelism (DSP), to significantly reduces communication overhead and improves throughput efficiency. DSP offers significant reductions in communication costs, and the experiments demonstrate its superiority over state-of-the-art sequence parallelism methods. In addition, the authors also successfully convinced reviewers by clarifying some concerns. Thus, acceptance is recommended.